# COVID-19 Awareness among Undergraduate Medical Students in Trinidad: A Cross-Sectional Study

**Srikanth Umakanthan** [1,*]**, Aalia Ramlagan** [2] **, Celine Ramlal** [2]**, Pavitra Ramlal** [2]**, Diva Ramlochan** [2]**, Anagha-Devi Ramlogan** [2]**, Priya Ramnarace** [2]**, Tanisha Ramnarine** [2] **and Aderlene Ramnath** [2]

1    Department of Para-Clinical Sciences, Faculty of Medical Sciences, The University of the West Indies, St. Augustine 685509, Trinidad and Tobago
2    Faculty of Medical Sciences, The University of the West Indies, St. Augustine 685509, Trinidad and Tobago
*    Correspondence: srikanth.umakanthan@sta.uwi.edu; Tel.: +1-(868)-4730728

**Abstract: Background**: The urgency for heightened levels of the Coronavirus disease of 2019 (COVID-19) awareness is due to their estimated face-to-face participation in the COVID-19 pandemic and similar pandemics. The unavailability of updated pandemic information is a significant challenge. There is no available data or previous studies undertaken to investigate the level of pandemic awareness of medical students in Trinidad, Tobago, or the wider Caribbean. **Methods**: A cross-sectional study of medical students, years one to five, at the University of the West Indies (UWI) St. Augustine campus, Faculty of Medical Sciences, was conducted using random sampling. Data was collected using a 20-item questionnaire structured to test awareness. Chi-square analysis was done using SPSS version 28.0.1.0 (142). **Results**: Of the 137 participants, 100% claimed to be aware of the COVID-19 pandemic, mainly via social media and the Ministry of Health press conferences. Though all claimed to be aware, 98.5% were aware of COVID-19 being a viral infection, whilst 87.6% were aware of the modes of transmission. Less than half of the population (45.3%) stated they were prepared to be a frontline worker exposed to and treating COVID-19 patients, while the majority (76.6%) were worried about exposure to the virus. **Conclusions**: The data collected in this research indicated that the level of awareness increases with higher levels of education, whereas age has no effect. Additionally, it was determined that undergraduate medical students had an average knowledge base of COVID-19 but would need training programs to increase their preparedness as future healthcare professionals. Lastly, it was discovered that the two top sources of information were social media and press conferences held by the government.

**Keywords:** COVID-19; awareness; transmission; vaccination; healthcare

## 1. Introduction

The Coronavirus disease of 2019 (COVID-19) pandemic has caused a high public health concern with the dominance of healthcare professionals and students to combat its high transmission rates. The exigency for enhancing awareness of COVID-19 among medical students is attributed to their estimated face-to-face or indirect participation in the COVID-19 pandemic and similar pandemics [1]. Additionally, the endorsement of medical students as campaigners for advancing health awareness during health crises, such as COVID-19, is due to the public perception of them being comparably more knowledgeable concerning health issues than the remaining youth populace [2,3]. Therefore, medical students must be provided with valid health awareness information for dissemination to their family, friends, and the broader society in their advocacy for precautionary action during pandemics [3,4].

In referencing the findings of a Jordanian study, it was unearthed that 9.7% of the total medical student population considered the wearing of a face mask as a precaution [5]. The findings of this study were completely different from that of a similar study carried out on

Chinese residents, which revealed that 98.0% of Chinese residents admitted to wearing a face mask [6]. It was further revealed in the Jordanian study that 87.0% of medical students considered handwashing, while 83.1% considered staying at home as a precautionary measure [5]. Equally important was the finding of a similar study on medical students in Iran, which unearthed the average rate of correct answers concerning the "personal disclosure of preventive behaviours and perception of risk" by medical students as 89.6% and 79.60%, respectively [7].

The awareness of medical students concerning modes of COVID-19 transmission is critical for maintaining their safety during volunteerism. Essential to note were the findings of a Jordanian study, which reflected confusion in the awareness of medical students regarding their knowledge of "airborne and respiratory droplets modes of transmission," as 90% of the students described "close contact" and exposure to "contaminated surfaces" as modes for transmission. In comparison, 41.8% of the population perceived the virus was airborne [8].

Additionally, a similar study in Kuwait revealed that more than 18% of medical students understood the "viral transmission risk" and considered staying away from infected persons a precautionary measure [9].

In their influential capacities, medical students are expected to function as "vectors" of medically valid information to reduce vaccine hesitancy in the community during health awareness campaigns [10,11]. A study carried out to assess the knowledge of medical students concerning the COVID-19 vaccine revealed that 18 % of participants out of a population of 1747 (pre-clinical) perceived vaccination as inefficient in decreasing the community spread, while approximately 42.1% of the population stated the significance of the vaccines in preventing the spread of COVID-19. Interestingly, the finding was that pre-clinical students, 67.1%of the population, perceived the vaccine as a potential mode of transmission. Additionally, 34.3% of the population admitted a knowledge deficiency concerning the effectiveness of the Pfizer vaccine. Also, 1/4 (411) of the population lacked knowledge of the vaccine's effect on immunity, and 67.3% of the respondents responded correctly to the vaccine's effect on immunity [12].

The unavailability of updated pandemic information is a significant challenge medical students face during COVID-19. Therefore, they must be provided with access to updated pandemic information to sustain clerkship in a rapidly evolving healthcare system during the COVID-19 pandemic [1,5,13,14]. A recent study unearthed that 83.44% of the total medical student population utilized social media to access COVID-19 information, while 84.8% relied on online search engines. It was revealed in the same study that 64.1% of the population relied less on medical search engines [5].

Medical students, being inclined to altruism, are considered "clinicians in training" and an "available resource" during health crises [15,16]. However, they are exposed to unnecessary risks, such as "moral trauma and adverse health outcomes," through the inadvertent exploits of strained healthcare systems during pandemics [15–17]. A study undertaken on medical students in the USA revealed that 70% of the participants felt prepared to participate in an emergency before commencing the elective of disaster preparedness medical school [17]. In comparison, 11% felt unprepared after training [18]. Since values such as "altruism, service in time of crisis, and solidarity with the profession" are reinforced in volunteerism, medical students must be equipped with the knowledge, skills, and attitudes for meaningful participation in the healthcare system during pandemics [18–20].

The transition of the medical education process from a face-to-face to a virtual setting during COVID-19 has been met with significant challenges [21]. It is significant to consider the inclusion of these challenges encountered during the present pandemic in discussions for a medical education reform initiative to ensure the "integrity and continuity" of medical education and training programs during future pandemics [21,22].

## 2. Materials and Methods

### 2.1. Aim

The study was aimed 1. to determine whether the level of awareness is based on age and level of education, and 2. to determine the extent of knowledge that undergraduate medical students of the University of the West Indies (UWI), St. Augustine, have on COVID 19, its effect on preparedness as future frontline workers, and the reliability of their sources of information.

### 2.2. Study Setting

A cross-sectional study grounded on disseminating a questionnaire via a Google form was executed at the UWI St. Augustine campus, Faculty of Medical Sciences, and MBBS on 10 May 2022. It was reviewed by the Ethics Committee on 5 May 2022. The study population consisted of medical students from years one to five. The sample size was calculated using a Raosoft online sample calculator. The sample size was 138, with a 95% confidence level and a 5% error margin. Population correction to study sample size was done using the Cochran formula, $n = n0/1 + (n0/N)$, giving a value of 125. Additionally, a 10% increase was done to account for the persons who did not partake in the study, resulting in 137. Using an online Google form, reasons why the participant was selected, what was expected of them, withdrawal from participation, publishing of data, and (online) informed consent was placed at the beginning. The questionnaire contained demographic and awareness-based questions. Email addresses were not taken during data collection via using the online Google form questionnaire. The survey responses were downloaded to an Excel spreadsheet. The data was cleaned and imported into IBM SPSS version 28.0.10 (142) for analysis.

### 2.3. Study Design and Population

A cross-sectional study was conducted to assess COVID-19 awareness among undergraduate medical students of the UWI St. Augustine campus. The study was carried out using an online Google form for the questionnaire (which was in English), as face-to-face was not ideal during the COVID-19 pandemic. Medical students from year one to year five from the UWI, St. Augustine campus, Faculty of Medical Sciences. Sample size justification for the proportion used for the initial sample size was taken from Ikhlaq et al. The awareness and attitude of the undergraduate medical students towards the 2019-novel Coronavirus had an expected prevalence of $> 90\% = 0.9$. The respective year's group representatives obtained the source of the MBBS population. The inclusion criteria were all MBBS students at the UWI St. Augustine campus, from year one to year five. Exclusion criteria were students repeating a year.

### 2.4. Measure Tools and Statistical Analysis

The questionnaire was drafted based on the most common variables that have produced a functional variance and outcome in the results during the detailed literature search with related original students. It was composed in English. The study was conducted on twenty medical students to validate the questionnaire. The reliability of the questionnaire was assessed using Cronbach's alpha test, which produced a reliability coefficient of 0.79, indicating a high internal consistency.

The questionnaire comprises four principal components: 1. demographic profile; 2. socio-economic awareness; 3. knowledge, attitude, and practice; and 4. source of knowledge on COVID-19. The questionnaire included 5 independent and 15 dependent variables and was disseminated randomly via an online Google form to the medical students of year one to year five in their respective class group chats. Data were collected electronically and analysed. Descriptive analysis and validation were done to report percentages using Statistical Package for Social Sciences (SPSS) IBM software version 25.0. A Chi-square test was done, and the asymptotic value was found to determine the significance between relevant variables. The variables for testing attitude and practice on COVID-19

were not graded on a point scale to avoid bias within the study results. The sample students were principally from years one and two, and the point-scale grading would have created more confusion during their answer input, thereby affecting the frequency and percentage results with a positive and negative attitude.

*2.5. Ethical Clearance*

Ethics approval was obtained from the campus Research Ethics Committee of the UWI St. Augustine campus on 5 May 2022. The committee reviewed and approved the study protocol, information on who was participating in this study, the online consent form, and the online questionnaire. Participants were fully informed about the nature of the study, why they were selected, what they were expected to do, withdraw from participation, and how the data was going to be used or published at the beginning of the online questionnaire. Those who chose to take part in this study had to grant permission for the data to be created from the answers given. They had to confirm that they were over 18 years of age. Only after clicking the permission box were they permitted to fill out the questionnaire. Additionally, information was provided (email address and contact number) to contact a counsellor if this study caused any psychological discomfort. Therefore, the ethical considerations for this study included informed consent, voluntary participation, anonymity, and no potential to cause harm.

**3. Results**

A total of 137 medical students filled out the questionnaire. Over 75.2% of the study participants were females. The participants' age ranged from 19–35 years. 92.7% of the participants were in Group I, and 7.23% were in Group II. Of the participants, 100% stated they were aware of the COVID-19 virus and its global outbreak. The study group included medical students from years one to five, with a high distribution in year one (33.6%) and year two (27%) among the study participants. The least were from year five (8.8%), owing to their hectic clinical posting during the COVID-19 pandemic. Table 1 displays this along with the respective number of participants in each year group.

**Table 1.** Demographics and responses of the participants, 2022.

| Characteristic | Participants (n = 137) No. (%) |
|:---:|:---:|
| Sex | |
| Male | 34 (24.8%) |
| Female | 103 (75.2%) |
| Age | |
| Group I (19–26) | 127 (92.7%) |
| Group II (27–35) | 10 (7.23%) |
| Year of Study | |
| Year 1 | 46 (33.6%) |
| Year 2 | 37 (27%) |
| Year 3 | 24 (17.5%) |
| Year 4 | 18 (13.1%) |
| Year 5 | 12 (8.8%) |

COVID-19: Coronavirus Disease of 2019.

Data analysis concentrating on the knowledge and source awareness about COVID-19 proved that social media (86.1%) and the Ministry of Health press conferences (62.8%) were the popular options for medical students. The Ministry of Health press conferences were conducted twice on weekdays and once during the weekend. Hence, the students and the general public were receiving frequent updates on COVID-19 through social media, television, and radio broadcasts. Newspaper information was the least awareness source (38%) owing to the severe pandemic restrictions imposed by the government. Table 2 holds information on the level of awareness of COVID-19 among medical students.

**Table 2.** Different sources used by participants to gather information on COVID-19.

| Questions | Participants (n = 137) No.% |
|---|---|
| 1. Are you aware of the COVID-19 Virus and its global outbreak? | |
| 1.1 Yes | 137 (100) |
| 1.2 No | 0 (0) |
| 2. As a medical student, how did you become aware of COVID-19? | |
| 2.1 Newspaper articles | 52 (38.0) |
| 2.2 Ministry of Health Website | 57 (41.6) |
| 2.3 Ministry of Health Press Conference | 86 (62.8) |
| 2.4 Friends and Family | 77 (56.2) |
| 2.5 Medical professionals | 66 (48.2) |
| 2.6 Social Media | |
| 2.7 Television | 81 (59.1) |

COVID-19: Coronavirus Disease of 2019.

Data survey and analysis concentrating on the attitude segment has been shown in Table 3 with relevant information and response percentages. The maximum correct responses were obtained for 1. COVID-19 being a viral infection (98.5%); 2. preventive measures of the COVID-19 spread (96.8%); 3. COVID-19 is transmitted through droplets (87.6%), and 4. primary symptoms of COVID-19 (75.2%). The least correct response was obtained for the survey question indicating the recommended waiting time period for an asymptomatic person with COVID-19 to be vaccinated (18.2%). Participants were aware of the emergency signs of COVID-19 (65.3%), the incubation period (25.5%), and knowledge that the vaccine does not prevent its spread (27%). Additionally, 45.3% felt prepared to be a frontline healthcare worker.

**Table 3.** Current level of awareness of medical students on COVID-19.

| Question (Correct Answer) | Participants (n = 137) No.% |
|---|---|
| 1. COVID-19 is a viral infection. | 98.5% |
| 2. COVID-19 is transmitted via droplets, close contact, touching contaminated surfaces, and staying in a poorly ventilated area or crowded indoor setting. | 87.6% |
| 3. Social distancing, hand washing, mask-wearing, avoiding social gatherings, disinfection, and avoidance. | 96.8% |
| 4. Fever, cough, fatigue, muscle aches, headaches, shortness of breath, diarrhoea, nausea/vomiting, loss of taste and smell, sore throat, and runny nose are the primary symptoms of COVID-19. | 75.2% |
| 5. Chest pain, trouble breathing, cyanosis, and the inability to stay awake are emergency signs of COVID-19. | 65.3% |
| 6. The incubation period for COVID-19 is 14 days. | 25.5% |
| 7. Vaccination does not prevent the spread of COVID-19. | 27% |
| 8. The recommended period of time for an asymptomatic/mildly symptomatic person with COVID-19 to wait before being vaccinated is 10 days after a positive result. | 18.2% |
| 9. The recommended period of time for individuals who had COVID-19 and were treated with monoclonal antibodies to wait before being vaccinated is 3 months. | 71.5% |
| 10. Based on your current awareness surrounding COVID-19, do you consider yourself prepared to be a frontline health worker treating potential COVID-19 patients? (Yes) | 45.3% |

COVID-19: Coronavirus Disease of 2019.

Table 4 shows the Chi-square test done for Awareness scores and Year Groups having an asymptotic significance of 0.371.

**Table 4.** Chi-Square Test for Year of Study and Awareness scores.

| | Value | df | Asymptotic Significance (2-Sided) |
|---|---|---|---|
| Pearson Chi-Square | 34.006 | 32 | 0.371 |
| Likelihood Ratio | 42.811 | 32 | 0.096 |
| N of Valid Cases | 137 | | |

The Chi-square test was done as a non-parametric statistic to level the measurement of the variables. Table 5 shows the Chi-square test done for the awareness scores and age having an asymptotic significance. The Pearson Chi-square was used to test the independent categorical variables, as addressed in the survey questionnaire. The frequency of asymptomatic significance was 0.006. The likelihood ratio was utilized to identify the "best" of the two nested models, and the asymptomatic significance was 0.489.

**Table 5.** Chi-Square Test for Age and Awareness scores.

|  | Value | df | Asymptotic Significance (2-Sided) |
|---|---|---|---|
| Pearson Chi-Square | 134.742 | 96 | 0.006 |
| Likelihood Ratio | 95.730 | 96 | 0.489 |
| Linear-by-Linear association | 1.830 | 1 | 0.176 |
| N of Valid cases | 137 |  |  |

## 4. Discussion

The COVID-19 pandemic has caused serious health concerns over a prolonged period. The virus originated in Wuhan, China, in December 2019 and has caused a global health threat by depleting and challenging the healthcare sector worldwide [23,24]. The patients suffering from COVID-19 presented with fever, breathlessness, headache, and dry cough, requiring hospitalization. The COVID-19 virus belongs to the family Coronoviridae and is an enveloped RNA virus composed of an envelope (E) protein, membrane (M) protein, and spike (S) protein [25]. The transmission mode includes droplet aerosols and close contact [26].

The public reliance on the source of information has been mainly on social media due to widespread lockdowns and quarantines over an extended period [27]. Social media has exhibited a broad spectrum of liability by providing accessible, quick-access information and causing the widespread transmission of unreliable and false information [28]. Following the unreliable authenticity of social media, the public has turned to the health care community to provide valuable and reliable information on COVID-19. Hence, medical students form a vital link between the public and the medical sector in providing valuable and practical resources on COVID-19 [29].

In this study, it was discovered that there was no significant difference between the age and awareness scores. This was evident by the asymptotic significance of 0.006 (Table 5) and the overall response of all the participants, regardless of age, selecting similar answers. This indicated that they possessed comparable levels of awareness of the COVID-19 virus. Hence, the first part of the hypothesis was rejected.

However, a significant difference was seen with both the year groups and the awareness scores. This is indicated by the asymptotic significance of 0.371 (Table 4) and the varying levels of knowledge among the year groups. Thus, the second part of the hypothesis was accepted. This reveals that COVID-19 awareness is not linked to age but instead to the level of education. It was expected that the higher years would have more awareness of COVID-19 and, thus, be more prepared. We expected these results due to the higher level and more years of experience in both knowledge and clinical training.

Only one year-5 student, two year-3, and two year-2 obtained a perfect awareness score of 9, while the majority achieved a score of 5, indicating moderate COVID-19 awareness. Based on Table 3, medical students demonstrated sufficient knowledge of COVID-19 transmission (87.6%), prevention (96.8%), primary symptoms (75.2%), and emergency signs (65.3%). However, few demonstrated adequate knowledge of the incubation period (25.5%) and vaccination (27%). It was unexpected that only 27% of medical students knew vaccination does not prevent transmission of COVID-19. While there is no current published literature in the region to compare our findings, the results were similar to Adil et al., Nemat et al., Hong J. et al., and Maheshwari S. et al. [2,4,13,15]. However, one main difference established when compared to the Ikhlaq study was that a higher percentage of participants, 65.1%, had better awareness of the incubation period for COVID-19 [1].

The results indicated that medical students are only moderately prepared, previously encountered in a self-evaluation where 54.7% stated they were not prepared to be frontline workers treating COVID-19 patients (Table 3). This supported a finding in the literature review where the students were deemed unprepared due to possessing limited knowledge [2,5,12,13,16,19,20]. Additionally, most students utilized social media as their main source of information; 86.1% (Table 2) of the participants were found to use social media as a primary source of information as well, following previous studies [5,9,12,16]. While social media has a vast expanse of information about COVID-19, is easily accessed, and is affordable, it contains misleading information [6]. Medical students must ensure that they verify the information used to educate the wider community to prevent and disprove the spread of incorrect data [8]. However, 62.8% (Table 2) of medical students viewed national press conferences, indicating that many students were informed with accurate information; this positively affects their awareness and preparedness as future medical professionals.

This study has possible limitations. The study design was cross-sectional, and this time where the study was conducted may not be representative. There was a lack of prior research studies on the topic in Trinidad and in the region. This meant no articles were available to gain a regional and national perspective. It was conducted among medical students; thus, these results cannot be generalized to healthcare professionals or the public. Moreover, articles utilized as references consisted of biased and unilateral views. The results may have inaccuracy as there is no guarantee that students did not use the internet to answer the questionnaire, skewing results.

Recommendations from our study analysis would include 1. The university can create an online platform with updates and articles from credible sources tailored to new diseases and global outbreaks, such as the COVID-19 pandemic, to prevent medical students from turning to social media as their primary data source. 2. The university can introduce pandemic preparedness into the medical curriculum or introduce a new course providing clinical training specific to COVID-19 for medical students to be better prepared as frontline workers if necessary. 3. More healthcare professionals and medical students willing to participate can research less familiar diseases to provide quality data to the broader population.

## 5. Conclusions

The data analysis of the survey results indicates that the undergraduate medical students of Trinidad have scholarly knowledge, heightened awareness, and a positive attitude toward COVID-19. The demographic analysis revealed that the senior undergraduate students possessed higher knowledge, attitude, and awareness levels than the year one and two students. The difference in knowledge and awareness was due to the level of clinical exposure received by the higher class of undergraduate students. The most used source of information was found to be social media. While more reliable sources should be utilized in general, it was found that official and reliable sources of information, such as national press conferences, were used. The awareness of commonly exposed information, such as transmission, social distancing, and symptoms, was high among the student population, but they lacked the intricately detailed awareness of COVID-19. Awareness and attitude can be improved by implementing constant updates on COVID-19 through intra-campus news services, providing timely research articles, and allowing them to serve as volunteers in COVID-19 health camps.

**Author Contributions:** S.U.—Concept and study design, methodology, formal analysis, writing—original draft, and review and editing of the final draft. A.R. (Aalia Ramlaganand), C.R., P.R. (Pavitra Ramlal), D.R., A.-D.R., P.R. (Priya Ramnarace), T.R. and A.R. (Aderlene Ramnath)—methodology, devising investigation tools, data collection, formal analysis, statistical analysis, and tabulation. The final version was reviewed and approved by all the authors. All authors have read and agreed to the published version of the manuscript.

**Funding:** No funding or grant was received for this study.

**Institutional Review Board Statement:** Ethics approval was obtained from the campus Research Ethics Committee of the University of the West Indies, St. Augustine campus, on 5 May 2022. The committee reviewed and approved the study protocol (CREC-SA.1243/11/2021), information on who was participating in this study, the online consent form, and the online questionnaire.

**Informed Consent Statement:** Participants were fully informed about the nature of the study, why they were selected, what they were expected to do, withdrawal from participation, and how the data was going to be used or published at the beginning of the online questionnaire.

**Data Availability Statement:** Data and copies of the questionnaire are available upon reasonable request to the corresponding author.

**Conflicts of Interest:** The authors declare no conflict of interest.

**Survey Questionnaire**

1. Please state your age.
2. What is your sex?

   Male ☐
   Female ☐

3. Are you aware of the COVID-19 virus and its global outbreak?

   Yes ☐
   No ☐

4. Select which Year in Medicine you are in currently.

   Year 1 ☐
   Year 2 ☐
   Year 3 ☐
   Year 4 ☐
   Year 5 ☐

5. As a Medical Student, how did you become aware of COVID-19?

   Newspaper Articles ☐
   Ministry of Health Website ☐
   Ministry of Health Press Conference ☐
   Friends and Family ☐
   Medical Professionals ☐
   Social Media ☐
   Television ☐

6. Do you believe that higher-year Medical students(years 3 to 5) have increased awareness as compared to lower-year medical students (years 1 to 2)?

   Yes ☐
   No ☐

7. Does awareness of the COVID-19 virus among Medical students of The UWI, St. Augustine Campus affect the community? (inclusive of relatives)

   Yes ☐
   No ☐

8. Covid-19 is a
   - bacterial infection ☐
   - viral infection ☐
   - parasitic infection
   - fungal infection

9. Are you aware of the modes of transmission of COVID-19?

    Yes ☐
    No ☐

10. Select the options which you believe to be modes of transmission for covid-19.
    - via droplets (sneezing, coughing) ☐
    - close contact ☐
    - poorly ventilated area ☐
    - crowded indoor setting ☐
    - touching contaminated surfaces and then touching the eyes, nose, or mouth without cleaning the hands. ☐
    - from mother to child through the placenta ☐
    - from mother to child through breastmilk ☐

11. What safety precautions should be taken in preventing the spread of COVID 19?
    - social distancing ☐
    - hand washing ☐
    - mask-wearing ☐
    - avoiding social gatherings/crowded areas ☐
    - disinfection ☐
    - avoiding touching your face ☐
    - if Other, state: ☐

12. Select the primary symptoms of covid-19.
    - fever and cough ☐
    - fatigue and muscles aches ☐
    - headaches ☐
    - shortness of breath ☐
    - diarrhoea and nausea/vomiting ☐
    - loss of taste and smell ☐
    - sore throat and runny nose ☐
    - Other, please state ☐

13. What are the emergency signs of COVID-19?
    - pain/pressure in the chest ☐
    - trouble breathing ☐
    - pale or bluish discoloration of lips, skin, and nails. ☐
    - inability to stay awake ☐
    - Other, please state . . . . ☐

14. What is the incubation period for Covid-19?

15. Do you believe that vaccination prevents the spread of Covid-19?

    Yes ☐
    No ☐

16. Do you believe vaccination against COVID-19 is important?

    Yes ☐
    No ☐
    If yes, why?

17. What is the recommended period of time for an asymptomatic/mildly symptomatic person with COVID-19 to wait before being vaccinated?
    - 10 days after the positive COVID-19 test ☐
    - one month ☐
    - three months ☐
    - waiting period is not important, an individual can get vaccinated during the active stages of the infection ☐

18. What is the recommended period of time for individuals who had COVID-19 and were treated with monoclonal antibodies to wait before being vaccinated?

    - 2 weeks ☐
    - One month ☐
    - three months ☐
    - no importance ☐

19. Based on your current awareness surrounding COVID-19, do you consider yourself prepared to be a frontline health worker treating potential covid-19 patients?

    Yes ☐
    No ☐

20. Are you worried about the exposure, you as a potential frontline health worker, will have to COVID-19?

    Yes ☐
    No ☐

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
