# Peer review of "COVID-19 Awareness among Undergraduate Medical Students in Trinidad: A Cross-Sectional Study"

_ime, doi:10.3390/ime1020006_

Round 1

Reviewer 1 Report

This is well written study however there are several concerns that must be addressed before this study can be considered for publication.

1. some references are missing (e.g. pg1, ln41; p2,ln84...)

2. Rather than writing formulas, please write the name of the method used for sample size calculation). 

3. Table 6. is redundant and should be deleted.

Reviewer 2 Report

Dear Authors,

the study lacks a related Independent variable and Dependent variable. in addition, the study's measure tool is unclear; please address more.

the result finding related to lower data should address more instead only presenting the data.

Thank you.

Round 2

Reviewer 2 Report

Dear Authors,

The study's revised manuscript has resolved my concerns; I had no other new issues.

Thank you.